Isolation, diversity, and antimicrobial activity of fungal endophytes from Rohdea chinensis (Baker) N.Tanaka (synonym Tupistra chinensis Baker) of Qinling Mountains, China

An Chao 1 2
Ma Saijian 1 2
Shi Xinwei 2 3
Xue Wenjiao x-wenjiao@163.com 1 2
Liu Chen 1 2
Ding Hao 1 2
1 Shaanxi Institute of Microbiology , Xi’an , Shaanxi , China
2 Engineering Center of QinLing Mountains Natural Products, Shaanxi Academy of Sciences , Xi’ an , Shaanxi , China
3 Shaanxi Institute of Botany,Xi’ an Botanical Garden , Xi’ an , Shaanxi , China
U’Ren Jana
Electronic publication date: 2020 Jun 17
Publication date: 2020
Volume: 8
Electronic Location ID: e9342
Received 2019 Aug 30; Accepted 2020 May 21
Copyright: ©2020 An et al.
Copyright year: 2020
Copyright holder: An et al.
License: This is an open access article distributed under the terms of the Creative Commons Attribution License, which permits unrestricted use, distribution, reproduction and adaptation in any medium and for any purpose provided that it is properly attributed. For attribution, the original author(s), title, publication source (PeerJ) and either DOI or URL of the article must be cited.
License URL: https://creativecommons.org/licenses/by/4.0/

Keywords: Endophytic fungi, Diversity, Antimicrobial activity, Tupistra chinensis Baker, Qinling mountain

Funding: National Natural Science Foundation of China 21576160 Science and Technology Research Project of Shaanxi Province Academy of Sciences 2018nk-01 2018k-09 Shaanxi Science and Technology Project 2017NY-192 2018NY-152 2019NY-209 This project was funded by the National Natural Science Foundation of China (21576160), Science and Technology Research Project of Shaanxi Province Academy of Sciences (2018nk-01, 2018k-09), Shaanxi Science and Technology Project (2017NY-192; 2018NY-152; 2019NY-209). The funders had no role in study design, data collection and analysis, decision to publish, or preparation of the manuscript.

==============================
Endophytic fungi have been emerged as fruitful resources for producing structurally fascinating and biologically active secondary metabolites. However, endophytic fungi from medicinal plants of Qinling Mountains–the most important natural climatic boundary between the subtropical and warm temperate zones of China with an astonishingly high level of biodiversity–have rarely been explored as potential sources of novel fungal species and active secondary metabolites. In this study, a total of 371 fungal colonies were successfully isolated from 510 tissue segments of the medicinal Tupistra chinensis Baker collected from Qinling Mountains, China. Roots of T. chinensis Baker are used as a folk medicine to ameliorate pharyngitis and treat rheumatic diseases. A total of 100 representative morphotype strains were identified according to ITS rDNA sequence analyses and were grouped into three phyla (Ascomycota, Basidiomycota, Mucoromycota), seven classes (Dothideomycetes, Sordariomycetes, Eurotiomycetes, Microbotryomycetes, Agaricomycetes, Leotiomycetes, Mortierellomycetes), and at least 35 genera. The genera of Collectotrichum (IF, 29.92%), Fusarium (IF, 8.36%), Aspergillus (IF, 8.09%), and Dactylonectria (IF, 5.39%) were most frequently isolated from the tissues of T. chinensis Baker. The Species Richness Index (S, 65) and the Shannon-Wiener Index (H′, 3.7914) indicated that T. chinensis Baker harbored abundant fungal resources. Moreover, five isolates were potential new taxa because of low similarity of ITS sequences ranged from 95.09%∼96.61%. Fifteen out of 100 endophytic fungal ethyl acetate extracts exhibited inhibitory activities against at least one pathogenic bacterium or fungus. Two important lead compounds produced by two stains (F8047 and F8075) with high antimicrobial activities were identified using high performance liquid chromatography (HPLC) and ultra-performance liquid chromatography-quadrupole-time of flight mass spectrometry (UPLC–QTOF MS) analyses. In addition, it was noteworthy that the strain F8001, which may be a potential new species, showed antimicrobial activity and should be investigated further. Overall, these results indicated that the endophytic fungi from T. chinensis Baker could be exploited as a novel source of bioactive compounds.

Introduction

Due to the increase in antibiotic resistance among pathogens, discovering novel antimicrobials is of high importance (Payne et al., 2007). However, after many decades of exploration it is increasingly difficult to discover novel bioactive metabolites from common environments (e.g., soils) (Jiang et al., 2018; Zhang et al., 2018; Lopes et al., 2016). Thus, in recent years the discovery of novel bioactive metabolites from extremophilic microorganisms inhabiting marine environments, desert soils, hot springs, and mangrove forests have attracted great interest (Rateb et al., 2011; Nicoletti & Vinale, 2018; Sayed et al., 2019). In addition, endophytic fungi that inhabit the interiors various plant tissues without causing disease (Petrini, 1991) have been increasingly identified as a source of novel metabolites. Medicinal plants harbor a great diversity of culturable endophytes (Tan et al., 2014; Tan et al., 2018; Gong et al., 2019), which have can produce structurally fascinating and biologically active secondary metabolites (Helaly, Thongbai & Stadler, 2018; Kumari et al., 2018; Leylaie & Zafari, 2018; Mousa et al., 2015).

The goal of this study was to isolate fungal endophytes from T. chinensis Baker of Qinling Mountains region and to screen these fungi for antimicrobial activities and identify compounds using high performance liquid chromatography (HPLC) and ultra-performance liquid chromatography-quadrupole-time of flight mass spectrometry (UPLC–QTOF MS) analyses. The Qinling Mountains (32°30′−34°45′N, 104°30′−112°45′E), which is mainly located in the south of Shaanxi province in central China, is the most important natural climatic boundary between the subtropical and warm temperate zones of China, and supports an astonishingly high biodiversity (Ma et al., 2018; Yuan et al., 2017; Huang et al., 2016; Wang et al., 2018). T. chinensis Baker grows in the valleys at altitudes of 500–2,400 m in Qinling Mountains region. The genus Tupistra (Liliaceae), which has 12 species in China, is an important biological resource used as a folk medicine (Xiang et al., 2016a). The dried rhizome of T. chinensis Baker is reputed to be used as Chinese folk medicine to ameliorate pharyngitis and treat rheumatic diseases (Xiang et al., 2016a; Xiang et al., 2016b). Previously, it was reported that T. chinensis Baker produced the antifungal compounds against Pseudoperonospora cubensis and Phytophthora infestans (Zhu et al., 2018). However, to date there is no report on the diversity and antimicrobial activity of endophytic fungi isolated from T. chinensis Baker.

Materials & Methods

Plant material

A total of twenty wild plants of T. chinensis Baker were collected from Yingpan town, Shaanxi province of China (33°49′22′′N, 109°6′11′′E, altitude, 1,180 m). The land we accessed was publicly owned and undeveloped. These plants were carefully dug up, placed in sterile sampling bag, labeled, immediately transported to the laboratory, and then placed them into a refrigerator (4 °C) as described previously (Tan et al., 2014). Endophyte isolation procedures were performed within 48 h of collection.

Fungal isolation and cultivation

The plant tissues were processed with the method described by Qin et al. (2009). In brief, the samples were thoroughly washed in running water for 30 min, followed by a ultrasonic cleaning (200 W, 10 min), and then air-dried for 2 h at room temperature. After drying, the plant samples were surface-sterilized with the protocol described by Tan et al. (2012) with minor modifications. Air-dried plant samples were surface-sterilized using sequential washes in 70% ethanol for 1 m, 2.5% NaClO2 for 2 m, 70% ethanol for 1 m. Following sterilization, leaves were rinsed three times in sterile distilled water. We divided the 20 plants into root and stem tissues, which were then excised into 510 segments of 1–2 mm length (root segments: n = 270; stem segments: n = 240). Segments were placed on a series of isolation media including potato dextrose agar (PDA), malt extract agar (MEA) and sabouraud agar (SDA) medium using 90 mm petri plates. Each isolation medium was amended with amikacin sulfate (100 U/mL) to prevent the growth of bacteria. Seven tissue segments were placed on each Petri dish (90 mm), which were then sealed with parafilm and incubated at 28 °C for one week. Emergent fungal colonies were isolated and purified in PDA medium for further identification and bioactive assays (see below). Pure isolates growing on PDA medium were photographed and the agar piece plugs with pure isolates were stored at −80 °C in a 20% glycerol solution in engineering center of QinLing Mountains natural products, Shaanxi provincial institute of microbiology.

Molecular identification and phylogenetic analyses

To obtain fungal mycelia, each pure isolate was cultivated on plates containing PDA medium at 28 °C for 7 days. Mycelia were removed from media using sterile pipette tips and then ground in liquid nitrogen for DNA extraction using the TaKaRa MiniBEST Bacteria Genomic DNA Extraction Kit (Dalian, China). Genomic DNA was then used as the template for PCR amplification of the nuclear ribosomal DNA internal transcribed spacer (ITS) using the universal primers ITS1 (5′-TCCGTAGGTGAACCTGCGG-3′) and ITS4 (5′-TCCTCCGCTTATTGATATGC-3′) according to the description by White et al. (1990) with minor modifications. The final reaction volume was 50 µL, containing 5.0 µL of 10 ×Taq buffers, 4.0 µL of 200 mmol/L dNTPs, 2.0 µL of each primer at 10 µM, 0.5µL of Ex Taq enzyme (TaKaRa, Dalian), and 5.0 µL of genomic DNA. PCR amplification was performed using TProfessional Standard 96 Gradient (Biometra, Germany) using the following cycling parameters: 1 min 95 °C; followed by 35 cycles of 15 s at 95 °C, 30 s at 55 ° C, and 1m at 72 °C; and a final 10 m extension at 72 °C. Five µL of each PCR product was analyzed electrophoretically in 1% (w/v) agarose gels stained with GelRed (Shanghai Generay Biotech Co., Ltd, China). The PCR products were subsequently purified and sequenced by BGI Biotechnology (Shenzheng, China). The raw obtained sequences were aligned using MEGA 5.05, edited manually, and then BLAST (Basic Local Alignment Search Tool) was used to search for the best match in the National Center for Biotechnology Information (NCBI) GenBank database (http://www.ncbi.nlm.nih.gov/) to identify endophytic fungi. Sequences with similarity over 97% belonged to the same genus. The sequences obtained in this study were submitted to GenBank database with accession numbers from MK367469 to MK367568. The evolutionary history was inferred as described by Wei et al. (2018) and Felsenstein (1985). All sequences were aligned by MEGA 5.05 using alignment prepared with Clustal W and all positions containing gaps and missing date were deleted. Finally, the maximum likelihood phylogenetic trees were constructed for each of families using MEGA software 5.05 (Stamatakis, 2006).

Crude extract preparation of fungal fermentation broth

Each isolate was cultured on PDA for 7 days, after which the plugs of each fungus were used to inoculate liquid cultures containing 250 mL Erlenmeyer flask containing 50 mL potato dextrose (PD) culture medium containing 200 g/L potato extract and 20 g/L dextrose. All isolates were incubated on a rotary shaker at 28 °C and 230 rpm for 14 days. The fermentation broth was collected by centrifugation at 8,000 rpm for 8 min. Fifty mL of fermentation filtrate was extracted with 50 mL ethyl acetate (three extractions total) and the organic phase was concentrated using a rotary evaporator on 50 °C water bath to remove organic solvent as described by Xing et al. (2011) with minor modifications. The crude extracts were diluted with pure methanol to 10 mg/mL and sterilized by filtration using an organic filter (0.22 µm, Shanyu Co., Ltd, China).

Antimicrobial activity

Using the agar diffusion method described by Wang et al. (2019), we screened the ethyl acetate crude extracts from fermentation filtrates of 100 fungal strains for antimicrobial activities against seven pathogens, including Bacillus cereus, Escherichia coli, Bacillus subtilis, Staphylococcus aureus, Pseudomonas aeruginosa, Xanthomonas oryzae pv. oryzae and Candida albicans. The detailed operation procedure are as follows: 10 mL culture of the C. albicans grown 2 days in Sabouraud liquid medium at 28 °C was added to the 200 mL of the Sabouraud agar medium, while 10 mL cultures of pathogenic bacteria grown 12 h in the Luria-Bertani (LB) liquid medium at 28 °C was added to the 100 mL of the LB agar medium, mixed gently, and then poured slowly on the petri dish (90 mm) used as the test plate. Six mm sterilized straws were used for perforating the plate, and the agar blocks were removed with sterilized toothpick. The 100 µL fermentation filtrate EtoAc extracts (10 mg/mL) were added to the hole of the test plate. The plates were incubated at 37 °C for 24 h for the pathogenic bacteria or at 28 °C for 48 h for C. albicans. Ampicillin sodium (1 mg/mL) and Actidione (1 mg/mL) were used as a positive antimicrobial controls and pure methanol was used as a negative control. Antimicrobial activities were evaluated by measuring the diameter of the inhibition zones. All experiments were replicated three times.

Identification of antimicrobial compounds by HPLC and UPLC-QTOF MS analyses

We identified antimicrobial compounds using HPLC and UPLC-QTOF MS with the procedure described by Tan et al. (2018) and Hu et al. (2016). In brief, ethyl acetate crude extracts were dissolved in 2 mL pure methanol and were filtered through 0.22-µm syringe filters prior to UPLC-QTOF MS analyses (WATERS I-Class VION IMS QTof). Chromatographic separation was performed with ACQUITY UPLC BEH C18 column (1.7 µm, 2.1 × 100 mm) with an injection volume 5µL and a binary gradient elution mixture comprising water with 0.1% formic acid (A) and 0.1% formic acid in methanol (B) as follows: 0–5.0 min, 5–50% B; 5.0–12.0 min, 50–100% B; 13.0–15.0, 100–5% B. The mobile phase was applied at a flow rate of 0.4 mL/min and the temperature of the column oven was set to 35 °C. The MS was operated in negative ion mode and was set to total ion chromatogram mode with the following mass conditions: capillary voltage = 1.0 kV, low collision energy = 6V, source temperature = 100 °C, desolvation temperature = 500 °C, and desolvation gas flow = 800 L/h. Data acquistition and processing were conducted using Masslynx version (Waters, Manchester, UK).

Diversity analyses of the endophytic fungi

The isolation rate (IR) was calculated using the formula: IR (%) = (Ni/Nt) ×100, Ni: the number of the segments isolated the fungal species, Nt: the total number of segments incubated. The isolation frequency (IF) represented the frequency of the occurrence of certain endophytic fungi in total isolates based on the number of isolates (N). The relative abundance (RA) was calculated based on the number of all isolates number (N). The diversity of fungal species from T. chinensis Baker was evaluated using the Species Richness Index (S) and Shanon–Weiner Index (H′) with the procedure described by Fedor & Spellerberg (2013). Species Richness Index (S) was obtained by counting the number of endophytic fungal species in corresponding plant tissues. The proportions of the endophytic fungi against each pathogen were calculated according to the number of strains with inhibitory activity for each pathogen, a total of 100 endophytic fungi were participated in antimicrobial activities assessment in this study.

Table 1 Isolation frequency (IF) of each endophytic fungal species from T. chinensis Baker.

Closest species	Root	Leaf	Total	
	PDA	SDA	MEA	Subtotal	PDA	SDA	MEA	Subtotal			
	N	IF	N	IF	N	IF	N	IF	N	IF	N	IF	N	IF	N	IF	N	IF	
Cladosporium cladosporioides	–	–	–	–	–	–	–	–	4	5.13	3	4.55	–	–	7	3.95	7	1.89	
Cladosporium cucumerinum	–	–	2	2.63	2	4.65	4	2.06	–	–	–	–	–	–	–	–	4	1.08	
Cladosporium sp	3	4.00	–	–	1	2.33	4	2.06	–	–	–	–	–	–	–	–	4	1.08	
Leptospora rubella	–	–	3	3.95	–	–	3	1.55	–	–	–	–	–	–	–	–	3	0.81	
Didymella pinodella	–	–	–	–	–	–	–	–	3	3.85	–	–	–	–	3	1.69	3	0.81	
Phoma bellidis	–	–	–	–	–	–	–	–	–	–	2	3.03	–	–	2	1.13	2	0.54	
Phoma sp	–	–	–	–	–	–	–	–	2	2.56	–	–	–	–	2	1.13	2	0.54	
Periconia byssoides	3	4.00	–	–	–	–	3	1.55	–	–	–	–	–	–	–	–	3	0.81	
Leptosphaeria sp	3	4.00	–	–	1	2.33	4	2.06	–	–	–	–	–	–	–	–	4	1.08	
Setophoma terrestris	–	–	–	–	–	–	–	–	–	–	2	3.03	–	–	2	1.13	2	0.54	
Setophaeosphaeria citricola	3	4.00	2	2.63	–	–	5	2.58	–	–	–	–	–	–	–	–	5	1.35	
Alternaria alternata	–	–	–	–	–	–	–	–	3	3.85	2	3.03	–	–	5	2.82	5	1.35	
Pleosporales sp	–	–	2	2.63	–	–	2	1.03	–	–	–	–	–	–	–	–	2	0.54	
Aspergillus flavipes	–	–	2	2.63	1	2.33	3	1.55	3	3.85	2	3.03	–	–	5	2.82	8	2.16	
Aspergillus flavus	–	–	–	–	–	–	–	–	–	–	–	–	2	6.06	2	1.13	2	0.54	
Aspergillus hiratsukae	2	2.67	–	–	–	–	2	1.03	–	–	–	–	–	–	–	–	2	0.54	
Aspergillus pseudoglaucus	–	–	–	–	–	–	–	–	–	–	2	3.03	3	9.09	5	2.82	5	1.35	
Aspergillus sp	2	2.67	3	3.95	–	–	5	2.58	–	–	2	3.03	2	6.06	4	2.26	9	2.43	
Aspergillus sydowii	2	2.67	2	2.63	–	–	4	2.06	–	–	–	–	–	–	–	–	4	1.08	
Penicillium chrysogenum	–	–	–	–	–	–	–	–	3	3.85	–	–	1	3.03	4	2.26	4	1.08	
Penicillium citrinum	–	–	2	2.63	–	–	2	1.03	–	–	–	–	–	–	–	–	2	0.54	
Penicillium oxalicum	2	2.67	–	–	–	–	2	1.03	2	2.56	–	–	–	–	2	1.13	4	1.08	
Penicillium sp	2	2.67	–	–	–	–	2	1.03	–	–	–	–	–	–	–	–	2	0.54	
Talaromyces funiculosus	–	–	–	–	–	–	–	–	–	–	–	–	2	6.06	2	1.13	2	0.54	
Merimbla ingelheimensis	3	4.00	–	–	–	–	3	1.55	3	3.85	–	–	–	–	3	1.69	6	1.62	
Leptostroma sp	2	2.67	–	–	–	–	2	1.03	–	–	–	–	–	–	–	–	2	0.54	
Phomopsis lactucae	–	–	–	–	–	–	–	–	3	3.85	–	–	2	6.06	5	2.82	5	1.35	
Colletotrichum acutatum	4	5.33	2	2.63	2	4.65	8	4.12	–	–	–	–	–	–	–	–	8	2.16	
Colletotrichum godetiae	–	–	–	–	–	–	–	–	5	6.41	4	6.06	1	3.03	10	5.65	10	2.70	
Colletotrichum liriopes	–	–	–	–	–	–	–	–	14	17.95	8	12.12	2	6.06	24	13.56	24	6.47	
Colletotrichum sp	9	12.00	6	7.89	3	6.98	18	9.28	–	–	3	4.55	2	6.06	5	2.82	23	6.20	
Colletotrichum truncatum	4	5.33	17	22.37	21	48.84	42	21.65	–	–	–	–	–	–	–	–	42	11.32	
Annulohypoxylon annulatum	4	5.33	–	–	–	–	4	2.06	–	–	–	–	–	–	–	–	4	1.08	
Mortierella alpina	–	–	–	–	–	–	–	–	–	–	3	4.55	–	–	3	1.69	3	0.81	
Hypoxylon begae	2	2.67	–	–	–	–	2	1.03	–	–	–	–	–	–	–	–	2	0.54	
Hypoxylon fragiforme	3	4.00	–	–	–	–	3	1.55	–	–	–	–	–	–	–	–	3	0.81	
Hypoxylon sp	–	–	2	2.63	–	–	2	1.03	–	–	–	–	–	–	–	–	2	0.54	
Trichoderma citrinoviride	–	–	–	–	–	–	–	–	3	3.85	–	–	–	–	3	1.69	3	0.81	
Trichoderma sp	–	–	–	–	–	–	–	–	–	–	3	4.55	–	–	3	1.69	3	0.81	
Cylindrocarpon olidum	–	–	–	–	–	–	–	–	3	3.85	–	–	–	–	3	1.69	3	0.81	
Cylindrocarpon sp	2	2.67	–	–	–	–	2	1.03	–	–	–	–	–	–	–	–	2	0.54	
Dothideomycetes sp	–	–	–	–	–	–	–	–	3	3.85	3	4.55	–	–	6	3.39	6	1.62	
Dactylonectria macrodidyma	–	–	–	–	–	–	–	–	7	8.97	10	15.15	3	9.09	20	11.30	20	5.39	
Fusarium oxysporum	3	4.00	–	–	–	–	3	1.55	2	2.56	3	4.55	5	15.15	10	5.65	13	3.50	
Fusarium proliferatum	–	–	–	–	–	–	–	–	3	3.85	–	–	–	–	3	1.69	3	0.81	
Fusarium sp	5	6.67	3	3.95	–	–	8	4.12	–	–	–	–	–	–	–	–	8	2.16	
Fusarium solani	–	–	–	–	–	–	–	–	3	3.85	2	3.03	2	6.06	7	3.95	7	1.89	
Nectria haematococca	–	–	–	–	–	–	–	–	3	3.85	–	–	–	–	3	1.69	3	0.81	
Stachybotrys echinata	–	–	3	3.95	–	–	3	1.55	–	–	–	–	–	–	–	–	3	0.81	
Chaetomium murorum	–	–	3	3.95	–	–	3	1.55	–	–	–	–	–	–	–	–	3	0.81	
Chaetomium nigricolor	–	–	–	–	–	–	–	–	–	–	–	–	2	6.06	2	1.13	2	0.54	
Chaetomium sp	–	–	–	–	–	–	–	–	–	–	3	4.55	–	–	3	1.69	3	0.81	
Coniochaeta sp	3	4.00	2	2.63	–	–	5	2.58	–	–	–	–	–	–	–	–	5	1.35	
Nigrospora oryzae	–	–	4	5.26	3	6.98	7	3.61	–	–	–	–	–	–	–	–	7	1.89	
Arthrinium camelliae-sinensis	–	–	–	–	7	16.28	7	3.61	–	–	–	–	–	–	–	–	7	1.89	
Arthrinium arundinis	–	–	2	2.63	–	–	2	1.03	–	–	–	–	–	–	–	–	2	0.54	
Arthrinium yunnanum	3	4.00	5	6.58	–	–	8	4.12	–	–	–	–	–	–	–	–	8	2.16	
Pestalotiopsis vismiae	–	–	2	2.63	–	–	2	1.03	–	–	–	–	–	–	–	–	2	0.54	
Biscogniauxia sp	3	4.00	–	–	–	–	3	1.55	–	–	–	–	–	–	–	–	3	0.81	
Nemania serpens	–	–	–	–	–	–	–	–	3	3.85	–	–	–	–	3	1.69	3	0.81	
Nemania sp	3	4.00	7	9.21	2	4.65	12	6.19	–	–	–	–	–	–	–	–	12	3.23	
Xylaria sp.	–	–	–	–	–	–	–	–	–	–	5	7.58	2	6.06	7	3.95	7	1.89	
Clitopilus sp.	–	–	–	–	–	–	–	–	3	3.85	–	–	–	–	3	1.69	3	0.81	
Peniophora cinerea	–	–	–	–	–	–	–	–	–	–	2	3.03	2	6.06	4	2.26	4	1.08	
Rhodotorula mucilaginosa	–	–	–	–	–	–	–	–	–	–	2	3.03	–	–	2	1.13	2	0.54	
Total	75	100	76	100	43	100	194	100	78	100	66	100	33	100	177	100	371	100	
Tissue numbers	90	90	90	270	80	80	80	240	510	
IF (%)	83.33	84.44	47.78	71.85	97.5	82.5	41.25	73.75	72.75	
Species richness (S)	36	35	65	
Shaanon–Wiener index (H′)	3.1402	3.2661	3.7914	

Results

Isolation, sequencing, and diversity of the endophytic fungi from T. chinensis

In this study, a total of 371 fungal colonies were successfully isolated from 510 tissue segments of T. chinensis Baker with three different isolation media (Table 1). The greatest number of endophytic fungi were isolated on the PDA medium (IR, 41.24%), followed by SDA medium (IR, 38.27%) and MEA medium (IR, 20.49%). In contrast, similar IR values were obtained for roots and shoots. The 371 isolates initially were assigned to 100 representative morphotypes according to their culture characteristics on PDA. ITS rDNA sequences subsequently were generated for a representative of each morphotype (File S1). Based on the sequence similarity threshold (SSA, 97%∼100%) 95 isolates were categorized at the genus level, while the remaining five isolates remained unidentified at the genus level. Phylogenetic analyses using maximum likelihood (File S2) identified 35 fungal genera representing three phyla (Ascomycota, Basidiomycota, Mucoromycota), seven classes (Dothideomycetes, Sordariomycetes, Eurotiomycetes, Microbotryomycetes, Agaricomycetes, Leotiomycetes, Mortierellomycetes), 15 orders, 24 families, 35 genera and 65 taxon (Fig. 1 & Table 1). The Species Richness Index (S) and Shannon-Wiener Index (H′), which are two important parameters for diversity analysis, were 65 and 3.7914 for T. chinensis Baker, respectively.

Figure 1 Relative abundance (RA, %) of endophytic fungi at the level of class (A, D, G). order (B, E, H), family (C, F, I).

Similar to previous studies using culture-based methods (Hamzah et al., 2018; Yao et al., 2016; Li et al., 2016), the majority of fungi isolated from T. chinensis Baker were identified as Ascomycota (RA, 96.77%), which represented 31 genera. In addition, three genera belonged to Basidiomycota (RA, 2.42%), and only one genus belonged to Mucoromycota (RA, 0.81%). Dothideomycetes, Sordariomycetes and Eurotiomycetes have been reported as the three dominant classes in previous studies of endophytic fungi (Li et al., 2016; Tan et al., 2018; Yao et al., 2016). In our present study, Sordariomycetes (RA, 70.89%) was most abundant class, followed by the Eurotiomycetes (RA, 13.48%) and Dothideomycetes (RA, 11.86%), which was agreement with the previous studies (Tan et al., 2018) (Fig. 1A). Fig. 1B presented the RA of endophytic fungi at the order level and Glomerellales (RA, 31.01%) was the most abundant community in this study. Glomerellaceae (RA, 29.92%), Nectriaceae (RA, 15.90%), Aspergillaceae (RA, 12.94%) were three most abundant families in this study as shown in Fig. 1C and 16 families including Coniochaetaceae, Mortierellaceae, Phaeosphaeriaceae, Trichocomaceae, Trichosphaeriaceae, Periconiaceae, Leptosphaeriaceae, Pleosporaceae, Aspergillaceae, Rhytismataceae, Valsaceae, Stachybotryaceae, Lasiosphaeriaceae, Trichosphaeriaceae, Sporocadaceae, Entolomataceae, Sporidiobolaceae have only one species of the endophytic fungi from T. chinensis Baker. In addition, the distribution of endophytic fungi varied in different tissues as shown in Figs. 1D– Figs. 1I).

In previous studies, Colletotrichum, Fusarium, Aspergillus were the dominant genera of endophytes in different host plants (Tan et al., 2018; Gong et al., 2019; Salazar-Cerezo et al., 2018) and similar result was obtained in this study (Colletotrichum (IF, 29.92%), Fusarium (IF, 8.36%), Aspergillus (IF, 8.09%)). In addition, fungi representing the genera Biscogniauxia, Leptosphaeria, Leptostroma, Annulohypoxylon, Stachybotrys, Didymella, Peniophora, Dactylonectria, Nectria, Peniophora, Rhodotorula, Setophaeosphaeria, Clitopilus were obtained in this study. Endophytes in these genera have been reported much less frequently in previous studies (Sritharan et al., 2019; Gong et al., 2019; Li et al., 2018).

Antimicrobial activity screening of the ethyl acetate extracts from endophytic fungal culture filtrates

As shown in Table 2, 15 out of 100 endophytic fungal ethyl acetate extracts (15%) showed inhibitory activity against at least one pathogenic bacterium or fungus (File S3). The other 85 extracts did not show inhibitory activities. The proportions of inhibitory activity against different pathogens were 7% (B. cereum), 6% (B. subtili s), 6% (S. aureus), 2% (P. aeruginosa), 4% (C. albicans), 1% (E. coli), and 4% (X. oryzae pv. oryzae), respectively. The relatively low proportion of activity against the Gram-negative bacterium such as E. coli (1%) and P. earuginosa (2%) were observed, which was in accordance with previous studies (Deshmukh, Verekar & Bhave, 2014).

Table 2 Antimicrobial activities of culturable endophytic fungi from T. chinensis Baker.

Isolates No	Taxa (Accession number)	Inhibition zone in diameter on Petri plate (mm)	
		B. cereus	E.coli	B. subtilis	S. aureus	P. aeruginosa	X. oryzae	C. albicans	
F8001	Leptospora sp. (MK367469)	19.3 ± 0.2	–	–	16.2 ± 0.4	11.5 ± 0.3	–	–	
F8002	Chaetomium sp. (MK367470)	–	–	–	11.1 ± 0.2	10.2 ± 0.3	–	–	
F8003	Leptostroma sp. (MK367471)	–	–	–	14.2 ± 0.2	–	–	17.2 ± 0.4	
F8032	Annulohypoxylon sp. (MK367499)	–	–	15.4 ± 0.3	–	–	–	–	
F8036	Biscogniauxia sp. (MK367503)	12.3 ± 0.2	–	14.3 ± 0.4	–	–	–	–	
F8038	Penicillium sp. (MK367505)	15.7 ± 0.3	–	16.3 ± 0.1	–	–	15.3 ± 0.4	–	
F8047	Nigrospora sp. (MK367514)	–	–	–	–	–	–	19.5 ± 0.3	
F8049	Nemania sp. (MK367516)	–	–	–	–	–	–	11.3 ± 0.2	
F8073	Cylindrocarpon sp. (MK367537)	12.3 ± 0.2	–	12.5 ± 0.3	12.6 ± 0.3	–	10.3 ± 0.4	–	
F8075	Clitopilus sp. (MK367539)	14.6 ± 0.3	–	17.4 ± 0.5	43.6 ± 0.8	–	25.7 ± 0.4	–	
F8076	Nectria sp. (MK367540)	–	–	12.3 ± 0.2	9.5 ± 0.3	–	–	–	
F8080	Chaetomium sp. (MK367544)	–	–	–	–	–	12.4 ± 0.3	–	
F8081	Fusarium sp. (MK367545)	17.5 ± 0.5		–	–	–	–	–	
F8086	Trichoderma sp. (MK367549)	–	–	–	–	–	–	20.6 ± 0.8	
F8106	Aspergillus sp. (MK367568)	16.5 ± 0.3	16.8 ± 0.5	–	–	–	–	–	
Total	15	7	1	6	6	2	4	4	

Figure 2 Representative base peak ion chromatograms of strain F8047 ethyl acetate extracts (A) and standard griseofulvin samples (B) from UPLC-QTOF MS analyses performed in negative ion mode (C) standard griseofulvin (D) the arrow indicates the molecular.

The isolated strains with antimicrobial activity belonged to genera of Leptospora, Chaetomium, Leptostroma, Annulohypoxylon, Biscogniauxia, Penicillium, Nigrospora, Nemania, Cylindrocarpon, Clitopilus, Fusarium, Trichoderma, Aspergillus, respectively (Table 2). Among of these strains, Nigrospora sp. F8047 and Clitopilus sp. F8075 exhibited high antimicrobial activity. The inhibitory zone diameter against C. albicans of ethyl acetate extracts from F8047 was 19.5  ±  0.3 mm and the inhibitory zone diameter against S. aureus of ethyl acetate extracts from F8075 reached 43.6 ±  0.8 mm.

Identification of antimicrobial compounds from the strains F8047 and F8075

We used HPLC and UPLC–QTOF MS to analyze the antimicrobial compounds in the ethyl acetate extracts of the strains Nigrospora sp. F8047 and Clitopilus sp. F8075. The result of HPLC analysis showed that one of compounds in the ethyl acetate extracts from this strain had a same retention time with the standard griseofulvin (Figs. 2A & 2B). The MS spectra also showed the peak of (M+H)+ at m/z 352.0789 (Fig. 2C) for the corresponding compound from this strain was similar with that for griseofulvin standard (m/z 352.0788) (Fig. 2D). The results suggested that griseofulvin was an antifungal activity compound in ethyl acetate extracts from the strain F8047. In previous studies, species of Nigrospora have been widely reported to produce griseofulvin (Rathod et al., 2014).

The antibacterial compound of ethyl acetate extracts from strain Clitopilus sp. F8075, which showed high inhibitory activity against S. aureus with the inhibitory zone diameter 43.6 ±0.8 mm, were identified by UPLC-QTOF MS analyses (Fig. 3A). The results showed that one of compounds in the extracts from this strain had a peak of ((M+H)+ at m/z 378.2482 (Fig. 3B), which was the similar molecular weight with the pleurotropin (m/z 378.5). We speculated that the pleurotropin, which had been reported to be produced by Clitopilus spp. (Bailey et al., 2016; De Mattos-Shipley, Foster & Bailey, 2017), was an antibacterial compound in ethyl acetate extracts of the strain F8075. Later experiments confirmed the speculation and pleuromutilin was purified from the fermentation broth of F8075 guided by UPLC-MS and antimicrobial activity (data not shown).

Figure 3 Representative base peak ion chromatograms of strain F8075 ethyl acetate extracts (A) and the arrow indicates the molecular ion of pleuromutilin at m/z 378.2482 9.

Discussion

The Qinling Mountains are rich in medicinal plant resources, many of which have been used as traditional Chinese medicines by the local people. In recent years, increasing attention has focused on biodiversity and pharmacological properties of medicinal plants of the Qinling Mountains; however, as far as we know, few studies have attempted to evaluate the diversity of endophytes associated with these valuable plants. (Li et al., 2016). In this study, we investigated the diversity of the culturable endophytic fungi from T. chinensis Baker, which is one of most popular medicinal plants in Qinling Mountains.

Our results indicated that the plants harbored endophytic fungi with high diversity, comparable to recent studies on other medicine plants (Li et al., 2016; Yao et al., 2016; Zhou, Shen & Hou, 2017; Tan et al., 2018; Chen et al., 2019b). In addition, we isolated five fungal strains that may represent new species, as sequences in the NCBI GenBank database had low sequence similarity (i.e., 95.09%∼96.61%) (Vu et al., 2019). Our results indicate that T. chinensis Baker could be exploited as a novel source of endophytic fungi with antimicrobial activities, and we believe that other medicinal plants in Qinling Mountains also have the potential to be valuable resources for endophytic fungi.

Although the roots of T. chinensis Baker have been used in the traditional formulations, we isolated endophytes from both the stem and the root. It is notable that 36 species were obtained from the roots and 35 species from the stem of the T. chinensis Baker. Only six species (Aspergillus flavipes, Aspergillus sp., Penicillium oxalicum, Merimbla ingelheimensis, Colletotrichum sp. and Fusarium oxysporum) were cultured from both root and stem tissues. The result implicated that different tissues of T. chinensis Baker harbored different endophytic fungal species, although species richness and Shannon–Wiener diversity index were similar for different tissues of the plants.

Endophytic fungi have emerged as fruitful resources for producing structurally fascinating and biologically active secondary metabolites (Sunil, Shilpa & Sarita, 2015; Sunil et al., 2018). In our research, ethyl acetate extracts of 15 endophytic fungi (15% of total screened) showed inhibitory activity against pathogenic microorganisms. These 15 isolates represent fungal genera previously reported to produce antimicrobial compounds (Ouyang et al., 2018; Kamdem et al., 2018; Yu et al., 2018; Deshmukh, Verekar & Bhave, 2014). However, new bioactive compounds are constantly being discovered from plant endophytes (Rateb et al., 2011; Sayed et al., 2019; Chen et al., 2019a). Thus, the fungi with antimicrobial activity obtained in this study still have the potential to produce new structures natural products. In addition, it is noteworthy that the strain F8001, which may be a potential new species, showed antifungal activity and should be further researched.

Conclusions

In this study, the diversity and antimicrobial activities of the endophytic fungi from T. chinensis Baker were investigated for the first time. Our results illustrate that T. chinensis Baker harbors abundant fungal endophytes representing a diversity of taxonomic affiliations, including potentially new species. We found that 15 out of 100 endophytic fungal ethyl acetate extracts exhibited inhibitory activities against at least one of the pathogenic microorganisms. Specifically, the strains F8047 and F8075 with high antimicrobial activities produced two important types of antibiotics compounds. In addition, it was noteworthy that the strain F8001, which may be a potential new species, showed antifungal activity and should be investigated further. Overall, these results indicated that the endophytic fungi from T. chinensis Baker could be exploited as a novel source of bioactive compounds.

Supplemental Information

File S1 Sequencing analysis

Click here for additional data file.

File S2 Maximum-likelihood phylogenic analyses

Click here for additional data file.

File S3 The raw data for antimicrobial activity

Click here for additional data file.

Additional Information and Declarations

Competing Interests

Author Contributions

DNA Deposition

Data Availability

The authors declare there are no competing interests.

Chao An conceived and designed the experiments, performed the experiments, analyzed the data, prepared figures and/or tables, authored or reviewed drafts of the paper, and approved the final draft.

Saijian Ma performed the experiments, authored or reviewed drafts of the paper, and approved the final draft.

Xinwei Shi conceived and designed the experiments, analyzed the data, authored or reviewed drafts of the paper, plant material collection, and approved the final draft.

Wenjiao Xue conceived and designed the experiments, analyzed the data, prepared figures and/or tables, authored or reviewed drafts of the paper, and approved the final draft.

Chen Liu performed the experiments, authored or reviewed drafts of the paper, and approved the final draft.

Hao Ding performed the experiments, prepared figures and/or tables, and approved the final draft.

The following information was supplied regarding the deposition of DNA sequences:

The data is available at GenBank: MK367469 to MK367568.

The following information was supplied regarding data availability:

The raw data for antimicrobial activity is available at Figshare:

Chao, An (2020): The raw data for antimicrobial activity (Endophytic fungi of Tupista chinensis Baker).xlsx. figshare. Dataset. https://doi.org/10.6084/m9.figshare.12380615.v1.

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
