# Peer review of "Isolation, diversity, and antimicrobial activity of fungal endophytes from Rohdea chinensis (Baker) N.Tanaka (synonym Tupistra chinensis Baker) of Qinling Mountains, China"

_PeerJ, doi:10.7717/peerj.9342_

## Round 0.1 · original submission · Major Revisions

Although the manuscript is on an important and interesting topics, the reviewers have raised a number of significant concerns regarding the lack of sufficient negative controls and unclear presentation of the results.

·

Basic reporting

Major points:
1. In general, there is a lack of description or reference of known species identified in this study. Only observation was provided in some cases with no explanation or further discussion (e.g. on line 181, it’s not clear why Sordariomycetes and Dothideomycetes were most abundant groups). How are the relative abundance of these microorganisms compared to the previously reported data in the same region if any? On line 164, what’s the implication of having these six species in both tissues? Is it surprising to see this result? It’s not clear what these discoveries mean. Although the major focus of the paper is to look for new bioactive substance, it’s important to discuss the biodiversity of endophytic fungi of this plant since authors claimed that this was the first time it had been done.
2. Authors only include the positive data in this manuscript. The negative results should also be reported. On line 199, the data from the other 85 extracts that did not show inhibitory activity are not included.

Minor points:
1. There are some small grammar mistakes like on line 79 “As far as we known” should read “As far as we know”.
2. Appropriate citations should be included for methodology used. For instance, on line 189, “Neighbourhood-joining method” was originally described in Saitou N, 1987, Mol Biol Evol.

Experimental design

Major points:
1. There is no control in the antimicrobial activity screening done by authors. It’s not clear whether the inhibitory activity is truly due to the extracts or due to other factors like contamination. The results from the three replicates are not clearly presented. It’s not clear whether the higher antibacterial activity from F8047 and F8075 is significant or not. How does the inhibitory activity of these extracts compare to the extracts of fungi that are known to produce antibiotics? This comparison is crucial to determine whether there are truly bioactive substances.
2. Authors should specify “The other statistical analyses” on line 147. The statistical analysis is not clearly presented with no p-value or statistics given (e.g. line 163, 167).

Minor points:
1. It’s not clear how antimicrobial activities were evaluated. How were the proportions on line 201 obtained? Is there a cutoff for the change of diameter?

Validity of the findings

Major points:
1. Most data were presented in tables rather than figures. The content in all three tables can be better visualized with bar plots and/or pie plots. (e.g. https://www.nature.com/articles/s41598-019-50602-5.pdf)

Minor points:
1. On line 257, “This result suggested that the T.chinensis Baker plant harbor a great diversity of endophytic fungi resources.” This is not an accurate statement without comparing with other plants. Having four new species doesn’t mean “a great diversity”.
2. On line 152, what’s the percentage of tissue segments with fungal colonies? What’s the distribution of 371 colonies among the collected samples?

Additional comments

Chao el at reported 371 endophytic fungal colonies extracted from T. chinensis Baker at Qinling mountain in China. Using previously published fungal isolation and cultivation techniques, 65 species of endophytic fungi were identified as endosymbiont with the plant. Among these strains, three new species were identified as well as new antibacterial compounds. Authors claimed that these findings would make fungi from T. chinensis Baker a potential novel source for new bioactive compounds.
The question of this study is well-formed but it lacks proper controls, raw data including negative results, sufficient discussion of the findings and detailed statistics. Therefore, I suggest a major revision.

Reviewer 2 ·

Basic reporting

Please see below. My complete review is included in the Validity of findings section

Experimental design

My complete review is included in the Validity of findings section

Validity of the findings

Manuscript. No.: #40649

Title: Isolation, diversity, and antimicrobial activity of culturable fungal endophytes of Tupistra chinensis Baker from Qinling Mountains, China

Authors: Chao An, Saijian Ma, Xinwei Shi, Wenjiao Xue, Chen Liu, Hao Ding

This study’s background information is good but I am not convinced the results/conclusions provided meets the study goals based on the data provided in this manuscript. Methods section also need more detail for better understanding of their experimental design.

Overall, the authors need to clarify the following points in order to justify their data, results and conclusions:

1) The phylogenetic analysis as shown in Figure 1 and 2 and all subsequent conclusions based on this as mentioned in the results section needs to be supported with better bootstrap values. At this point, the bootstrap values are unacceptable to draw such conclusions.

2) The authors need to provide more details on how the antimicrobial activity was measured – on what criteria the zone diameters were measured, how the proportions of inhibitory activity were measured.
Please see additional comments/ suggestions below.

Introduction:
Lines 63-62 – Please elaborate what you mean by this and provide a reference.
Lines 66-68 – This sentence needs to be rephrased for better understanding of the readers. What do you mean by ‘excavation’?
Lines 67-69 – Sentence ‘For this reason…” needs references.

Materials and methods:
Lines 84-88 – Plant material – Authors need to provide the following information- a) how many plants were collected? b) How many of these were used for fungal isolation? c) How were the collected plants transported to the laboratory- under what conditions?
You mention some of these between lines 234-235 in discussions, this needs to be clearly included in methods.

Line 94 – please elaborate what ‘modifications’ were used.
Lines 95 – Expand what you mean by PDA, MEA and SDA medium, abbreviations need to be expanded and explained on first use.
Line 101 – At what conditions were the pure isolates stored? Please explain.
Line 119 – What is ‘levers’?
Line 128 – what is r? Is this rpm or rcf?
Line 131 – what do you mean by a ‘cupboard’?
Line 133 - please elaborate what ‘modifications’ were used.
Line 136- 141 – Antimicrobial activity – this section needs to much more detailed. What were the standards used for diameter measurements to determine antimicrobial activity? How were the proportion of inhibitory activity calculated?
Line 147 – What do you mean by ‘other statistical analyses’? Please elaborate.

RESULTS:
Line 165 – it is unclear what ‘observed simultaneously in two plant tissues’ means here.

Figure 1 – The bootstrap values displayed in this phylogenetic tree is unacceptable at this point. Your bootstrap values are too low for most of the phylogenetic clades to confirm statistical significance of this tree, and the subsequent conclusions displayed under the results section.
Although this is not the only one, for your benefit, I am sharing one reference paper here: https://academic.oup.com/sysbio/article-abstract/42/2/182/1730933/.

Figure 2 – (Lines 189 – 192) – In figure 2, the bootstrap values supporting this specific clade is unacceptable to conclude this.

Lines 197 – 211 – Authors need to further elaborate on how exactly these inhibition zones were traced, and using what standards zones of inhibition were measured. Authors also need to elaborate on how the proportion of inhibitory activity were calculated.

Additional comments

My complete review is included in the Validity of findings section

Reviewer 3 ·

Basic reporting

1. In this manuscript, the authors have isolated and characterized the endophytic fungi of Tupistra chinensis Baker from Qinling Mountains, China. Though the work has its scientific value, however the manuscript in the current form suffers from various flaws. So following are my comments/suggestions.
The manuscript suffers from a significant linguistic problem.
For example:
i. The genetic and metabolite levels of the endophytes have changed owing to resisting the stress from their habitats.
ii. The result showed that the PDA medium isolated more endophytic fungal (153, IR 41.24%), followed by SDA medium (142, IR 38.27%) and MEA medium (76, IR 20.49%).
iii. Five genera with higher isolation frequency (IF) were Phyllosticta spp (IF, 21.65%), Colletotrichum spp (IF, 13.40%), Arthrinium spp (IF, 8.76%), Aspergillus spp (IF, 7.22%), and Nemania spp (IF, 6.19%) in the root tissues, respectively. Nevertheless, those higher isolation frequency genera were Colletotrichum spp (IF, 22.03%), Dactylonectria spp (IF, 11.30%), Fusarium spp (IF, 11.29%), Aspergillus spp (IF, 9.03%),Cladosporium spp (IF, 3.95%), in the stem tissues, respectively.
iv. The composition and distribution of endophytic fungi exhibited significantly different in different host plants (Yuan et al., 2011; Jia et al., 2016).
v. Though most genera of the 15 endophyte funguses, which were isolated from of other ecological environment, have been reported to produce antibacterial compounds.
Besides these examples, there are such linguistic errors at several places. Thus, the English language must be significantly improved to ensure the clear delivery of your message to the readers. Besides, authors must ensure consistency throughout (eg. Use either fungi or funguses).
2. The introduction section lacks the motivation for the work carried out by the authors. The authors must provide sufficient literature in the Introduction section to highlight the gaps in this field of research in the current scenario. Besides, there is redundancy in their Introduction and Results section. For example, authors have provided the literature about the bioactive metabolites, steroids etc., however, in the current work, they are “just” checking the antimicrobial activity of the endophytic fungi. In this direction, the authors must add more literature.
Importantly, authors must discuss their results in parallel to in the Introduction section.

Experimental design

1. Here authors have explored the fungal diversity of a Tupistra chinensis Baker from Qinling Mountains, China. However, there is no novelty as authors have identified neither some novel fungal strains nor some novel bioactive compounds. Authors have made very superficial conclusions/speculations only and no validation was carried out for these speculations, which make this a shallow study. Such shallow investigations do not qualify the standards and the scope of PeerJ.
2. Research questions are not well defined as there is a lack of literature support to display the gap which authors have attempted to fulfill.
3. The molecular identification and phylogenetic analyses carried out by authors seem to be very shallow and lack rigorous investigation. Below are some points which authors must address:
4. There are no details of the taxonomic assignment and phylogenetic analysis in the manuscript. What were the cutoff values of ITS similarity with reference sequences for species, genus, order, family, and phyla delineation and reference(s) for these cutoff values? How have authors selected members for phylogenetic analysis? How have the aligned the sequences? Have they performed bootstrapping or not?
5. Authors have not provided the method details and reference of filtering paper method for antimicrobial assays.

Validity of the findings

1. Here authors have explored the fungal diversity of a Tupistra chinensis Baker from Qinling Mountains, China. However, there is no novelty as authors have identified neither some novel fungal strains nor some novel bioactive compounds. Authors have made very superficial conclusions/speculations only and no validation was carried out for these speculations, which make this a shallow study.
2. As mentioned above, the molecular identification and phylogenetic analyses carried out by authors seem to be very shallow and lack rigorous investigation.
3. Conclusions do not match the objectives mentioned by authors.

Additional comments

1. What are the isolation rate and isolation frequencies?
2. Authors have provided scientific names of few plants and common English names of few others. I recommend that authors should provide the scientific name and well as the common English name of the plants mentioned in the manuscript.
3. There are lots of errors in scientific names. Authors must correct these names. (For example, P.earuginosa to P. aeruginosa.; C.albicans to C. albicans)
4. The scientific names mentioned for the first time in the manuscript should be expanded.
5. The authors must adhere to the nomenclature, for example, Penicillium sp to Penicillium sp.
6. You have mentioned in the paragraph “As presented in Table 2, the strains isolated from T.chinensis Baker were grouped into two 174 phylum (Ascomycota (IF, 97.57%) and Zygomycota (IF, 2.43%))”. However, in Table 2, this are only two phyla viz., Ascomycota and Basidiomycota.
7. The paragraph “As presented in Table 2, the strains isolated from T.chinensis Baker were grouped into two phylum” is redundant. The results are already appended in Table 2. So, remove this section.
8. Authors have claimed that “As shown in Figure 2, the 190 results showed that the isolates F8001 were identified only at the genus level belonged to Leptospora spp and showed distinct clades with the its closest relatives species, which suggested that F8001 may be a potential new species of Leptospora spp.” Here, F8001 is clustering with Pleosporales sp. 38. So, how do you suggest it as a novel species of the genus Leptospora but not as a novel genus itself in the order Pleosporales.
9. Authors speculated that F8067 and F8079 are new species of the genus Nemania. What do you reason behind this hypothesis as the genus Nemania is itself scattered in the tree?
10. Authors have mentioned that the relatively low proportion of activity against the Gram-negative bacterium such as E.coli (2%) and P.earuginosa (2%) were observed. R. solanacearum is another Gram-negative bacterium yet showing 5% activity.
11. Authors have mentioned that “Based on the Ouyang et al. (2018) research, a new antibacterial compound named molicellins O, which was isolated from the endophytic fungus Chaetomium sp. Edf-10, exhibited antibacterial antibacterial activities against Staphylococcus aureus ATCC 29210”. But original paper cited here has not bacterial strain Staphylococcus aureus ATCC 29210. Clarify it.
12. The discussion part includes majorly the literature without any relevant relation to the authors’ work. It seems that authors are supporting the literature with their work, whereas they are supposed to support their work with literature. If there is not enough discussion then authors must merge the Results and Discussion section.
13. The conclusion is very shallow.
14. References are not as per the required format
For example:
i. Bailey AM, Alberti F, Kilaru S, Collins CM, de Mattos-Shipley K, Hartley AJ, Hayes P, Griffin A, Lazarus CM, Cox RJ, Willis CL, O'Dwyer K, Spence DW, Foster GD. 2016. Identification and manipulation of the pleuromutilin gene cluster from Clitopilus passeckerianus for increased rapid antibiotic production. Scientific reports 6: 25202-25212 DOI 10.1038/srep25202. (Scientific report to Scientific Reports).
ii. Cheng Z, Xu W, Liu L, Li S, Yuan W, Luo Z, Zhang J, Cheng Y, Li Q. 2018. Peniginsengins B⁻E, New Farnesylcyclohexenones from the Deep Sea-Derived Fungus Penicillium sp. YPGA11.Marine drugs 16(10):358-368 DOI 10.3390/md16100358. (Marine drugs to Marine Drugs).

---

## Round 0.2 · Minor Revisions

The reviewers have commented mostly favorably on the changes you have made to the manuscript, including the better structure and additional details that provide key information about your study. However, the manuscript still contains a number of grammatical issues and other issues. You will find that the reviewers have made a number of helpful suggestions to help guide your revisions, in particular those of R4.

·

Basic reporting

The revised manuscript is better structured.
In the revised manuscript, authors provide a better description of the scientific question and a deeper discussion about the impact of this study.

Experimental design

Proper controls are clearly explained in the revised manuscript.
The methods include more detail to replicate the experiments.

Validity of the findings

The detailed analysis of data is provided in the revised manuscript.
Supplementary data are provided.

Additional comments

Higher resolution images should be used for Figures.
The statistical analyses section in the results (line 162) does not include any hypothesis testing. It's more like taxonomic analysis.
Acknowledgements should not include funding statement.

Reviewer 2 ·

Basic reporting

The authors have made all changes to this manuscript based on my previous suggestions, and the manuscript is more articulate now. The English is a lot clearer, and overall the paper looks great now after all the revisions.

Experimental design

see above for my comments.

Validity of the findings

see above for my comments.

Additional comments

Thank you for making the changes to the manuscript based on my previous suggestions.

Reviewer 4 ·

Basic reporting

Line numbers are those in the review PDF

There are many grammatical / language errors throughout. I suggest to have a native English speaker edit. References for methods are often to the most recent publication rather than the original. I suggest to cite the originals for clarity. The raw data for constructing the phylogeny and the phylogeny itself are not made public - please make public. The sequence data in GenBank has the wrong host species listed - please fix.

Abstract
- There are a few grammatical / language mistakes:
- line 27: ... segments of T. chinensis Baker plant... (i.e., plant is awkward)
- line 30: numbers < 10 should be spelled out
- lines 33-34: ... were most frequently from the tissues...
- line 38: ... one of the pathogens microorganism.

Introduction
- There are several grammatical / language mistakes:
- lines 48-49: ... discovery of new structure active ingredient...
- line 64: ... endophytic fungi... is scarcely explored...
- line 56: Citations here are not ordered by year within author.
- line 66: It would be nice to have some taxonomy beyond genus described for the plant host - such as family.
- line 83: I recommend to avoid beginning sentences with numbers, and to write them out instead.

Experimental design

Many details of the culturing, extractions, and assaying could be explained further, for reproducibility. No real research question was defined. While the work is novel in that it is the first study of endophytes from this particular host, no new fungi nor metabolites were described. I suggest to describe the unknown fungi and confirm the identity of the metabolites described.

Materials and methods
Plant material
- lines 84-86: The citation Tan et al. 2014 describes collection somewhat differently than reported here, in that they first transported plants to the lab and then placed them into a refrigerator. I suspect the order of '... packed into a refrigerator (4 degrees C) and immediately transported back to the lab...' is reversed. If not, please describe in further detail the instrument / refrigerator used to transport the plants to the lab.
Fungal isolation and cultivation
- lines 89, 95: Which parts of the plant were fungi isolated from? Where did the 510 segments come from with respect to the 20 plants collected?
- lines 98-99: What size Petri plates were used, and how many tissue segments were plated on each plate? Were plates sealed with Parafilm, or otherwise contained?
Molecular ID and phylogenetic analysis
- There is a discrepancy between reporting only the forward primer used in White et al. (1990) and the PCR recipe reported that contains 2 µL of each primer.
- line 112: The word preceding 'Standard Gradient' appears misspelled.
- line 120: GenBank is misspelled, and the accessions have the wrong plant host species listed.
Antimicrobial activity
- lines 134-135: Italics are used incorrectly in some cases: 'pathogenic bacterial' should not be italicized, whereas oryzae (following pv.) should be. I see that you refer to 'pathogenic bacterial' again on line 145, but urge you to highlight this group otherwise, if even necessary, to avoid confusion with the latin nomenclature here.
- What size Petri dishes were used throughout?
- line 147: Is the use of methanol as a negative control appropriate? Wouldn't using ethyl acetate, or water, be more appropriate given the compounds being assayed were extracted using ethyl acetate?
- I think more needs to be described regarding how the ethyl acetate extracts were produced, beyond citing Wang et al. 2019 that has little additional details in that regard. For example, how were cultures grown prior to extraction? How much volume or biomass was extracted and using how much solvent? Please elaborate for clarity and transparency.
ID of compounds
- Much of the text in this section appears nearly identical to that in the citation Tan et al. (2018). Also, that citation cites another paper, "Fu, Y. J. et al. An analytical pipeline to compare and characterise the anthocyanin antioxidant activities of purple sweet potato cultivars. Food Chemistry 194, 46–54 (2016)", for some of the analysis in this section.
Statistical analyses
- line 171: 'The procedure' is the Shannon-Weiner identity and Tan et al. (2018) is not the original citation - I recommend to define it here, and/or use the original citation.
Results
- line 191: The citation here should also be reported during the method's first mention in the methods section
- Figure 1: The inclusion of only one taxon per genus compromises your ability to classify things beyond genus - you need to include sequences from other species in each genus in order to see where your sequences are placed in that context. However, I see that your longest sequence is around 650 bp, which is quite short. It is not acceptable to infer a phylogeny across this diversity using only the ITS region.
- Figure 2: I recommend to use colors and/or patterns that are more clearly distinguishable. Also, consider using stacked bar plots, rather than pie charts, for clarity.
- line 198: By different plants, do you mean among the 20 individuals you sampled? Please clarify.
- line 221: Leptosporium is misspelled

Validity of the findings

While classification to genus of the fungi sequenced, I suspect that they are not an accurate representation of all isolated. The estimation of isolation frequencies among the taxa reported are skewed due to certain isolates being assigned incorrectly to species. Phylogenetic placement of sequences using non-model-based inference did not include an inclusive sample of taxa within each genus such to provide additional insight beyond BLAST.

Sequence data at NCBI: Upon checking several accessions from this study, I notice that all have Disporopsis fuscopicta listed as the host plant species, as well as an unpublished study focused on that species listed as the publication source.

The multiple sequence alignment and resultant phylogeny are not accessioned or made public.

Molecular ID and phylogenetic analysis
- line 121: How many sequences were included in your phylogenetic analysis? What was the length of your multiple sequence alignment? If the number of sequences is close to the number of positions, constructing a de novo phylogeny may not be appropriate. Also, I recommend to use an inference method that uses an explicit model of sequence evolution, such as maximum likelihood or Bayesian methods. RAxML or MrBayes are good tools to use. Finally, please accession your alignment and phylogeny someplace where we can access them.

- Table 1: The taxon names here include species and often many within a single genus. I recommend to only classify taxonomy based on ITS sequences to the genus level. The phylogenetic analysis does not support classifying to species level because you did not include multiple species within each genus. The isolation frequencies are not accurate because of this. More importantly, many fungi that appear the same are often not, and some that look distinct can be the same (i.e., at the sequence level). I suspect that by assigning morphotypes and sequencing a single representative of each, that your conclusions based on that subsample do not accurately represent the fungal diversity in your entire culture collection.
- Table 3: Nice table! However I do not think you should classify your sequences past genus based on these data.
- Table 4: Also a cool table. However, work in foliar endophytes has shown species to diverge at ca. 95% ITS sequence similarity, so you are right on the line here, especially for some of the groups reported here (U'Ren et al. 2009 - Diversity and evolutionary origins of fungi associated with seeds of a neotropical pioneer tree - case study for analysing fungal environmental samples). I would hesitate to conclude that these are novel taxa without also stating that performing a multi-locus phylogenetic, or phylogenomic analysis with other members for each group would provide better insight, given the variability of ITS sequences even within a single species (and in some cases even a single spore).

Additional comments

Line numbers are those in the review PDF

In the abstract and introduction sections below, I highlight few instances of grammatical / language mistakes, but stop after that due to their consistent presence in the manuscript. I suggest the authors have a native english speaker go through the draft for clarity.

Abstract
- There are a few grammatical / language mistakes:
- line 27: ... segments of T. chinensis Baker plant... (i.e., plant is awkward)
- line 30: numbers < 10 should be spelled out
- lines 33-34: ... were most frequently from the tissues...
- line 38: ... one of the pathogens microorganism.
- lines 34-36: What are the values of species richness and diversity being compared to, such that they are 'high'?

Introduction
- There are several grammatical / language mistakes:
- lines 48-49: ... discovery of new structure active ingredient...
- line 64: ... endophytic fungi... is scarcely explored...
- line 56: Citations here are not ordered by year within author.
- line 66: It would be nice to have some taxonomy beyond genus described for the plant host - such as family.
- line 83: I recommend to avoid beginning sentences with numbers, and to write them out instead.

Materials and methods
Plant material
- lines 84-86: The citation Tan et al. 2014 describes collection somewhat differently than reported here, in that they first transported plants to the lab and then placed them into a refrigerator. I suspect the order of '... packed into a refrigerator (4 degrees C) and immediately transported back to the lab...' is reversed. If not, please describe in further detail the instrument / refrigerator used to transport the plants to the lab.
Fungal isolation and cultivation
- lines 89, 95: Which parts of the plant were fungi isolated from? Where did the 510 segments come from with respect to the 20 plants collected?
- lines 98-99: What size Petri plates were used, and how many tissue segments were plated on each plate? Were plates sealed with Parafilm, or otherwise contained?
Molecular ID and phylogenetic analysis
- There is a discrepancy between reporting only the forward primer used in White et al. (1990) and the PCR recipe reported that contains 2 µL of each primer.
- line 112: The word preceding 'Standard Gradient' appears misspelled.
- line 120: GenBank is misspelled, and the accessions have the wrong plant host species listed.
- line 121: How many sequences were included in your phylogenetic analysis? What was the length of your multiple sequence alignment? If the number of sequences is close to the number of positions, constructing a de novo phylogeny may not be appropriate. Also, I recommend to use an inference method that uses an explicit model of sequence evolution, such as maximum likelihood or Bayesian methods. RAxML or MrBayes are good tools to use. Finally, please accession your alignment and phylogeny someplace where we can access them.
Antimicrobial activity
- lines 134-135: Italics are used incorrectly in some cases: 'pathogenic bacterial' should not be italicized, whereas oryzae (following pv.) should be. I see that you refer to 'pathogenic bacterial' again on line 145, but urge you to highlight this group otherwise, if even necessary, to avoid confusion with the latin nomenclature here.
- What size Petri dishes were used throughout?
- line 147: Is the use of methanol as a negative control appropriate? Wouldn't using ethyl acetate, or water, be more appropriate given the compounds being assayed were extracted using ethyl acetate?
- I think more needs to be described regarding how the ethyl acetate extracts were produced, beyond citing Wang et al. 2019 that has little additional details in that regard. For example, how were cultures grown prior to extraction? How much volume or biomass was extracted and using how much solvent? Please elaborate for clarity and transparency.
ID of compounds
- Much of the text in this section appears nearly identical to that in the citation Tan et al. (2018). Also, that citation cites another paper, "Fu, Y. J. et al. An analytical pipeline to compare and characterise the anthocyanin antioxidant activities of purple sweet potato cultivars. Food Chemistry 194, 46–54 (2016)", for some of the analysis in this section.
Statistical analyses
- line 171: 'The procedure' is the Shannon-Weiner identity and Tan et al. (2018) is not the original citation - I recommend to define it here, and/or use the original citation.
Results
- line 191: The citation here should also be reported during the method's first mention in the methods section
- Figure 1: The inclusion of only one taxon per genus compromises your ability to classify things beyond genus - you need to include sequences from other species in each genus in order to see where your sequences are placed in that context. However, I see that your longest sequence is around 650 bp, which is quite short. It is not acceptable to infer a phylogeny across this diversity using only the ITS region.
- Figure 2: I recommend to use colors and/or patterns that are more clearly distinguishable. Also, consider using stacked bar plots, rather than pie charts, for clarity.
- line 198: By different plants, do you mean among the 20 individuals you sampled? Please clarify.
- line 221: Leptosporium is misspelled
- Table 1: The taxon names here include species and often many within a single genus. I recommend to only classify taxonomy based on ITS sequences to the genus level. The phylogenetic analysis does not support classifying to species level because you did not include multiple species within each genus. The isolation frequencies are not accurate because of this. More importantly, many fungi that appear the same are often not, and some that look distinct can be the same (i.e., at the sequence level). I suspect that by assigning morphotypes and sequencing a single representative of each, that your conclusions based on that subsample do not accurately represent the fungal diversity in your entire culture collection.
- Table 3: Nice table! However I do not think you should classify your sequences past genus based on these data.
- Table 4: Also a cool table. However, work in foliar endophytes has shown species to diverge at ca. 95% ITS sequence similarity, so you are right on the line here, especially for some of the groups reported here (U'Ren et al. 2009 - Diversity and evolutionary origins of fungi associated with seeds of a neotropical pioneer tree - case study for analysing fungal environmental samples). I would hesitate to conclude that these are novel taxa without also stating that performing a multi-locus phylogenetic, or phylogenomic analysis with other members for each group would provide better insight, given the variability of ITS sequences even within a single species (and in some cases even a single spore).

---

## Round 0.3 · Minor Revisions

I have reviewed your manuscript and while it is much improved I found it to still be missing key methodological details and to have significant issues with the clarity of the writing. Please see my specific comments and suggestions in the tracked version of your manuscript (which I will email you separately).

---

## Round 0.4 · accepted · Accept

Thank you for addressing all of the reviewers comments and suggestions.